# Observed variations in U.S. frost timing linked to atmospheric circulation patterns

Courtenay Strong[1] & Gregory J. McCabe[2]

Several studies document lengthening of the frost-free season within the conterminous United States (U.S.) over the past century, and report trends in spring and fall frost timing that could stem from hemispheric warming. In the absence of warming, theory and case studies link anomalous frost timing to atmospheric circulation anomalies. However, recent efforts to relate a century of observed changes in U.S. frost timing to various atmospheric circulations yielded only modest correlations, leaving the relative importance of circulation and warming unclear. Here, we objectively partition the U.S. into four regions and uncover atmospheric circulations that account for 25–48% of spring and fall-frost timing. These circulations appear responsive to historical warming, and they consistently account for more frost timing variability than hemispheric or regional temperature indices. Reliable projections of future variations in growing season length depend on the fidelity of these circulation patterns in global climate models.

[1] Department of Atmospheric Sciences, University of Utah, 135 S 1460 E, Salt Lake City, Utah 84112-0110, USA. [2] U.S. Geological Survey, Denver Federal Center, MS 412, Denver, Colorado 80225, USA. Correspondence and requests for materials should be addressed to C.S. (email: court.strong@utah.edu).

Changes and trends in the timing of seasonal transitions, such as the start, end and duration of the frost-free season are important effects and indicators of climate change[1,2]. Even though the timing and duration of the frost-free season is important for natural and managed environments, including agricultural production and the phenologies of plant and animal species[3–5], the relative importance of climatic factors that drive changes in components of the frost-free season over the past century is not well understood.

It is generally recognized that global warming results in an increase in daily minimum temperatures and subsequently an earlier onset of spring and a delayed end to the frost-free period, resulting in a longer growing season[1–8]. In contrast to these radiatively induced effects, forecasters have for decades recognized dynamically induced effects, whereby frost-triggering temperature extremes are associated with excitement of atmospheric circulation patterns conducive to delivery of anomalously cold-air masses[9], and this is supported by numerous case studies[10] and regionally focused composite analyses[11]. Frost timing is thus responsive to radiatively and dynamically induced temperature variations, and intertwining of these signals and mechanisms in the historical record[12,13] contributes to challenges in understanding their respective impacts on growing season length.

The timing of frost or seasonal transitions in specific regions is modestly correlated with teleconnections such as the Pacific Decadal Oscillation (PDO)[14], the Northern Annular Mode (NAM)[15] and the Atlantic Multidecadal Oscillation (AMO)[8]. Studies focused on regions of the western United States (U.S.) have reported correlations between frost timing and the Pacific-North American (PNA) pattern[16,17], which is characterized in its positive polarity by troughing over the eastern Pacific, ridging over the Rocky Mountains and troughing over eastern North America[18]. However, recent work extending frost-timing analysis to a century of data with coverage over the conterminous U.S. found statistically significant linkages to only the AMO[5]. The roles and relative importance of large-scale warming and dynamically induced atmospheric circulation effects in regulating observed variations in U.S. frost timing are thus not clear.

In this study, objective-clustering algorithms and optimization techniques were used to discover the atmospheric circulations that regulate conterminous U.S. frost timing for fall and spring during the period 1920–2012. The importance of these circulation patterns relative to effects from large-scale warming and regional seasonal mean temperature variability was then assessed via statistical modelling. We find that these circulations appear responsive to historical warming, and they consistently account for more frost timing variability than hemispheric or regional temperature indices. The finding that circulation consistently accounts for more frost timing variance than hemispheric annual mean or regional seasonal mean temperature was robust to the number of regions used in the clustering algorithm. Our results indicate that reliable projections of future variations in growing season length depend on the fidelity of the uncovered circulation patterns in global climate models.

## Results

**Frost timing spatial and temporal patterns**. In the conterminous U.S., the day of year with last spring frost (S) varied from ~90 in the southeastern U.S. to ~140 in the northern states (Fig. 1a). The day of year with first fall frost (F) varied from ~260 in the northern states to ~310 in the southeastern U.S. (Fig. 1b). Objective clustering of the spring frost timing data (detailed in Methods) partitioned the conterminous U.S. into four regions used in the analysis of spring (Fig. 1c), and the same objective clustering

of the fall frost timing data partitioned the conterminous U.S. into four regions used in the analysis of fall (Fig. 1d). The regions were similar between the two seasons, meaning both had a western region with the area east of the Rocky Mountains divided into a northern, central and southern region (Fig. 1c,d). Because the two sets of regions had similar boundaries, we use the same region naming convention for both seasons (North, Central, South and West) as indicated in Fig. 1c,d.

Within the region set for spring, area-weighted mean S in the North region ($S_N$) transitioned over the record toward smaller values indicating earlier last spring frost, with a $-0.8$ day decade$^{-1}$ trend from 1920 to 2012 (Fig. 2a). Similar changes occurred in the other analysis regions (Fig. 2a–d), with S in the Central region ($S_C$) trending at $-1$ days decade$^{-1}$, S in the South region ($S_S$) trending at $-0.5$ days decade$^{-1}$ and S in the West region ($S_W$) trending at $-0.6$ days decade$^{-1}$. All four trends are significant at the 95% confidence level as indicated by solid lines in Fig. 2a–d. The fraction of S variance accounted for by trend in the North and Central regions was 0.16 and 0.18, respectively, with smaller fractions in the South and West regions (0.05 and 0.08, respectively).

Within the region set for fall, F trended toward larger values indicating later first fall frost. F in the North region ($F_N$) underwent a 0.7 days decade$^{-1}$ trend with much of the change happening since 1990 (Fig. 2e), while no significant trend was observed for F in the Central region ($F_C$) or South region ($F_S$) (Fig. 2f,g). F in the West region ($F_S$) trended toward later timing at 0.5 days decade$^{-1}$ trend (Fig. 2h). The fraction of F accounted for by trend was 0.14 for the North region and 0.07 for the West region. Overall, the trend in the length of the frost-free season (F minus S) amounted to an increase of 13.3 days for the North region, 8.6 days for the Central region, 7.7 days for the South region and 10.7 days for the West region.

**Frost-timing relationship with temperature**. Beginning with larger scales conventionally used to summarize greenhouse gas-driven warming, we consider Northern Hemisphere annual mean temperature ($T_H$). The S indices had statistically significant negative correlation with $T_H$ in all four regions, accounting for 8–16% of the variance depending on region (Fig. 2a–d; Table 1). The F indices had statistically significant positive correlation with $T_H$ in the North, South and West regions, accounting for up to 27% of the variance depending on region (Fig. 2e–h; Table 1). Similar correlations were obtained between spring hemispheric mean temperature and S, and between fall hemispheric mean temperature and F (Supplementary Table 1). Replacing hemispheric mean temperature ($T_H$) with global mean temperature ($T_G$) tended to decrease the average correlation slightly (Supplementary Table 1).

To consider the role of regional temperature variability, we calculated spring (March–May) temperature indices for each of the four spring analysis regions (North, Central, South and West; Fig. 1c), and calculated fall (September–November) temperature indices for each of the four fall analysis regions (North, Central, South, West; Fig. 1d), providing a total of eight indices collectively denoted by $T_R$. $T_R$ accounted for more variance than $T_H$ in all cases except $S_N$, $S_C$ and $F_N$ (Table 1). Regional seasonal mean temperature was an especially reliable predictor of spring-frost timing in the West region (44% of variance), possibly because the continental divide partially shelters it from continental polar air masses and its climate is modulated by slowly varying sea-surface temperatures to the west[14,19].

**Spring frost timing relationship with circulation**. The geopotential height of the 500-hPa isobaric surface (Z) is often used to identify atmospheric teleconnections[20] (large-scale patterns featuring correlated pressure and circulation anomalies), and we

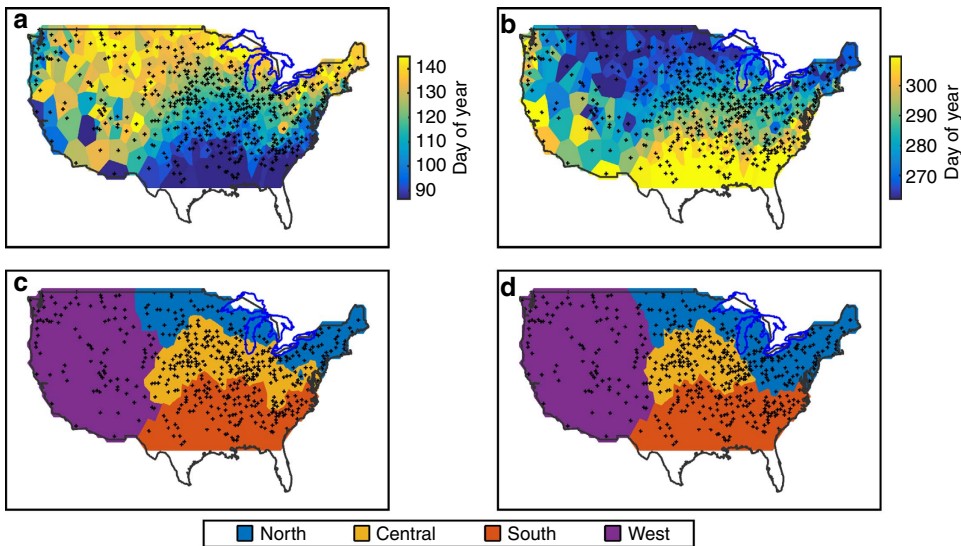

**Figure 1 | Frost timing spatial patterns.** (**a**) Time-average day of year with last spring frost. (**b**) Time-average day of year with first fall frost. (**c**) Objective clustering of stations into four regions based on day of year with last spring frost. (**d**) Objective clustering of stations into four regions based on day of year with first fall frost. In all panels, black dots indicate the locations of the 523 GHCN sites.

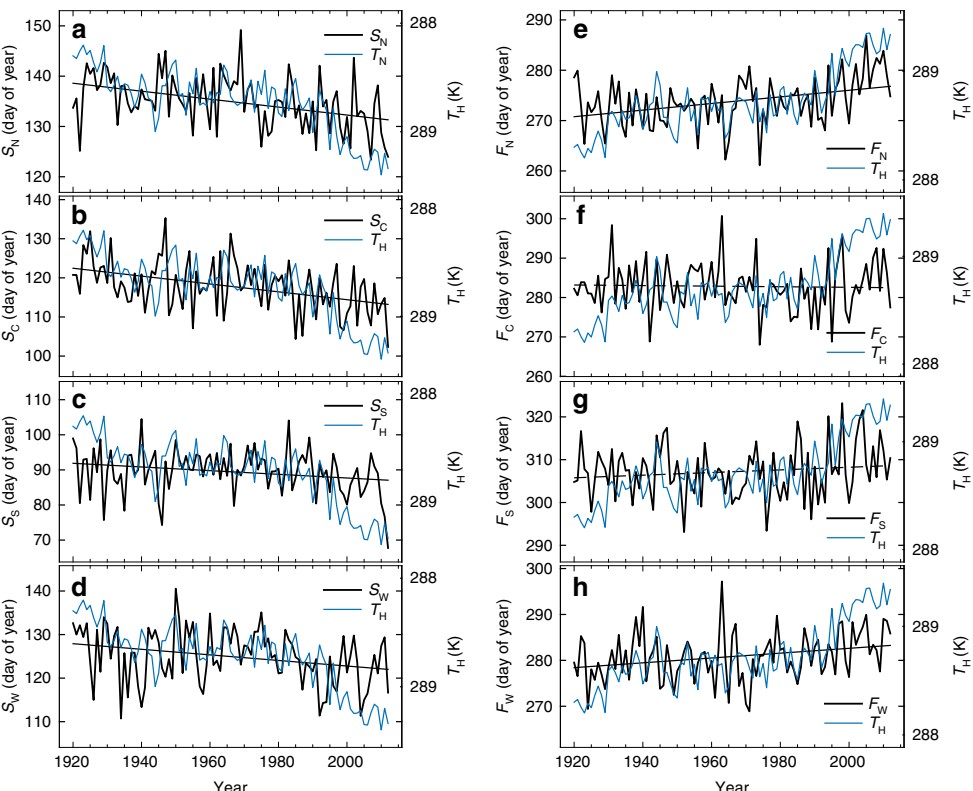

**Figure 2 | Temporal variations in frost timing.** (**a–d**) Day of year with last spring frost in the North region ($S_N$), the Central region ($S_C$), the South region ($S_S$) and the West region ($S_W$), shown with hemispheric annual mean near-surface air temperature ($T_H$). Note that the ordinate is reversed for $T_H$ to facilitate comparison to the $S$ indices. (**e–h**) Day of year with first fall frost in the North region ($F_N$), the Central region ($F_C$), the South region ($F_S$) and the West region ($F_W$), each shown with $T_H$. Trend lines are shown in each panel indicating presence (solid) or lack (dashed) of statistical significance at the 95% confidence level.

averaged $Z$ at each grid point over March through May ($Z_{MAM}$) to capture the months spanned by the spring frost timing index $S$. Beginning with the spring season North region, the correlation between $Z_{MAM}$ at each grid point and $S_N$ featured a tripole

pattern whose signs logically relate anomalously late spring frost to troughing over eastern North America conducive to cold-air delivery into the North region (Fig. 3a). If the circulation tripole in Fig. 3a is in fact a teleconnection in the atmosphere, we should

**Table 1 | Squared correlation between indices of frost timing, temperature and circulation, with bold indicating statistical significance and asterisk indicating negative correlation.**

| | Spring (March–May) | | | | Fall (September–November) | | | |
|---|---|---|---|---|---|---|---|---|
| | **North** | **Central** | **South** | **West** | **North** | **Central** | **South** | **West** |
| | $S_N$[a] | $S_C$[b] | $S_S$[c] | $S_W$[d] | $F_N$[e] | $F_C$[f] | $F_S$[g] | $F_W$[h] |
| $T_H$[i] | **0.16*(0.10)** | **0.14*(0.06)** | **0.09*(0.04)** | **0.08*(0.05)** | **0.27(0.02)** | 0.01(0.00) | **0.05(0.00)** | **0.17(0.01)** |
| $T_R$[j] | **0.06*(0.02)** | **0.06*(0.02)** | **0.27*(0.01)** | **0.44*(0.00)** | **0.16(0.04)** | **0.16(0.03)** | **0.20(0.02)** | **0.26(0.03)** |
| $C_{xx}$[k] | **0.32** | **0.26** | **0.38** | **0.48** | **0.41** | **0.25** | **0.38** | **0.29** |
| | $C_{SN}$[l] | $C_{SC}$[m] | $C_{SS}$[n] | $C_{SW}$[o] | $C_{FN}$[p] | $C_{FC}$[q] | $C_{FS}$[r] | $C_{FW}$[s] |
| $T_H$[i] | **0.06*** | **0.11*** | **0.06*** | 0.03* | **0.40** | **0.05** | **0.10** | **0.37** |
| $T_R$[j] | **0.42*** | **0.51*** | **0.55*** | **0.84*** | **0.15** | **0.27** | **0.31** | **0.44** |
| PNAI[t] | 0.03 | 0.02 | 0.19 | **0.55*** | **0.21*** | 0.03* | **0.59*** | 0.18 |

Numbers in parentheses indicate the fraction of residual frost index variance accounted for by the temperature index after the effect of $C_{xx}$ was linearly removed.
[a]Day of year with last spring frost in the North region.
[b]Day of year with last spring frost in the Central region.
[c]Day of year with last spring frost in the South region.
[d]Day of year with last spring frost in the West region.
[e]Day of year with first fall frost in the North region.
[f]Day of year with first fall frost in the Central region.
[g]Day of year with first fall frost in the South region.
[h]Day of year with first fall frost in the West region.
[i]Northern Hemisphere annual mean temperature.
[j]Regional mean temperature for spring or fall according to column heading.
[k]Circulation index corresponding to frost timing index in the column heading, meaning $C_{SN}$ for $S_N$, $C_{SC}$ for $S_C$ and so on.
[l]Circulation pattern corresponding to $S_N$.
[m]Circulation pattern corresponding to $S_C$.
[n]Circulation pattern corresponding to $S_S$.
[o]Circulation pattern corresponding to $S_W$.
[p]Circulation pattern corresponding to $F_N$.
[q]Circulation pattern corresponding to $F_C$.
[r]Circulation pattern corresponding to $F_S$.
[s]Circulation pattern corresponding to $F_W$.
[t]Pacific–North American Pattern Index.

be able to recover its spatial pattern and index time series via principal component (PC) analysis[21] of $Z_{MAM}$ as done in the classic studies that documented the PNA[22]. PCs are domain dependent[23], and PC time series that correlate significantly with $S_N$ can be found by constructing various reasonable analysis domains (spherical quadrangles) that enclose circulation anomalies in Fig. 3a. To optimize this relationship, we found the quadrangle PC analysis domain boundary that maximized the correlation between $S_N$ and one of the first three PC time series of $Z_{MAM}$ (algorithm detailed in Methods). Optimizing for $S_N$ yielded the grey box in Fig. 3a, with a high correlation between $S_N$ and the second PC of $Z_{MAM}$ which we denote by $C_{SN}$ to reflect its relation to $S_N$. Confirming that the optimization algorithm did not hone in on an arbitrary domain lacking physical meaning, the spatial coefficient or 'loading' pattern of the $C_{SN}$ mode aligns well with the tripole correlation pattern, featuring a ridge-trough structure over North America conducive to anomalously cold-air delivery into the North region (compare black contours to shading, Fig. 3a). The $C_{SN}$ time series was well correlated with $S_N$, accounting for 32% of its variance (Fig. 3b; Table 1). While the tripole pattern in Fig. 3a is broadly PNA-like in that it suggests a waveform teleconnection spanning from the North Pacific across North America, its centres of action are in antiphase with those of the PNA (compare black contours to small black and white circles, Fig. 3a), and $C_{SN}$ is thus not correlated with the PNA index (Table 1).

For the Central region, the correlation between $Z_{MAM}$ and $S_C$ revealed a similar tripole circulation pattern as found above for $S_N$, but shifted to the south as we would expect with the Central region being south of the North region (Fig. 3c). Optimizing for $S_C$ yielded the grey box in Fig. 3c, with a high correlation between $S_C$ and the second PC of $Z_{MAM}$ which we denote by $C_{SC}$ to reflect its relation to $S_C$. The spatial pattern of the $C_{SC}$ mode closely resembles the tripole correlation pattern featuring a ridge-trough structure over North America conducive to anomalously cold-air delivery into the Central region (compare black contours to

shading, Fig. 3c), and the $C_{SC}$ time series was correlated with $S_C$, accounting for 26% of its variance (Fig. 3d; Table 1). The $C_{SC}$ mode is in antiphase with the PNA (compare black contours to small black and white circles, Fig. 3c), and $C_{SC}$ is not correlated with the PNA index (Table 1).

For the South region, the correlation between $Z_{MAM}$ and $S_S$ revealed a similar tripole circulation pattern as found above for $S_N$ and $S_C$, but shifted south in alignment with the South region (Fig. 3e). Optimizing for $S_S$ yielded the grey box in Fig. 3e, with a high correlation between $S_S$ and the third PC of $Z_{MAM}$ which we denote by $C_{SS}$ to reflect its relation to $S_S$. The spatial pattern of the $C_{SS}$ mode closely resembles the tripole correlation pattern featuring a ridge-trough structure over North America conducive to anomalously cold-air delivery into the South region (compare black contours to shading, Fig. 3e), and the $C_{SS}$ time series was well correlated with $S_S$, accounting for 38% of its variance (Fig. 3f; Table 1). The $C_{SS}$ mode is nearly in antiphase with the PNA (compare black contours to small black and white circles, Fig. 3e), and $C_{SS}$ is not correlated with the PNA index (Table 1).

For the West region, the pattern of correlation between $Z_{MAM}$ and $S_W$ resembled the negative polarity of the PNA, featuring ridging over the eastern North Pacific indicating a weakened Aleutian low, downstream troughing conducive to cold-air delivery to the West region, and an additional centre of action near the Gulf of Mexico (compare shading to small black and white circles, Fig. 3g). The spatial pattern of the first PC of $Z_{MAM}$ within the optimized grey box in Fig. 3g ($C_{SW}$) captured this PNA-like circulation tripole well (compare black contours to shading, Fig. 3g), and the $C_{SW}$ time series accounted for 48% of the variance in $S_W$ (Fig. 3h; Table 1). $C_{SW}$ was also significantly negatively correlated with the PNA index ($r^2 = 0.55$; Table 1).

For comparison, we calculated spring frost timing correlations with indices of the PNA, North Atlantic Oscillation, Atlantic Multidecadal Oscillation and Southern Oscillation, finding correlations smaller than the circulation indices presented above

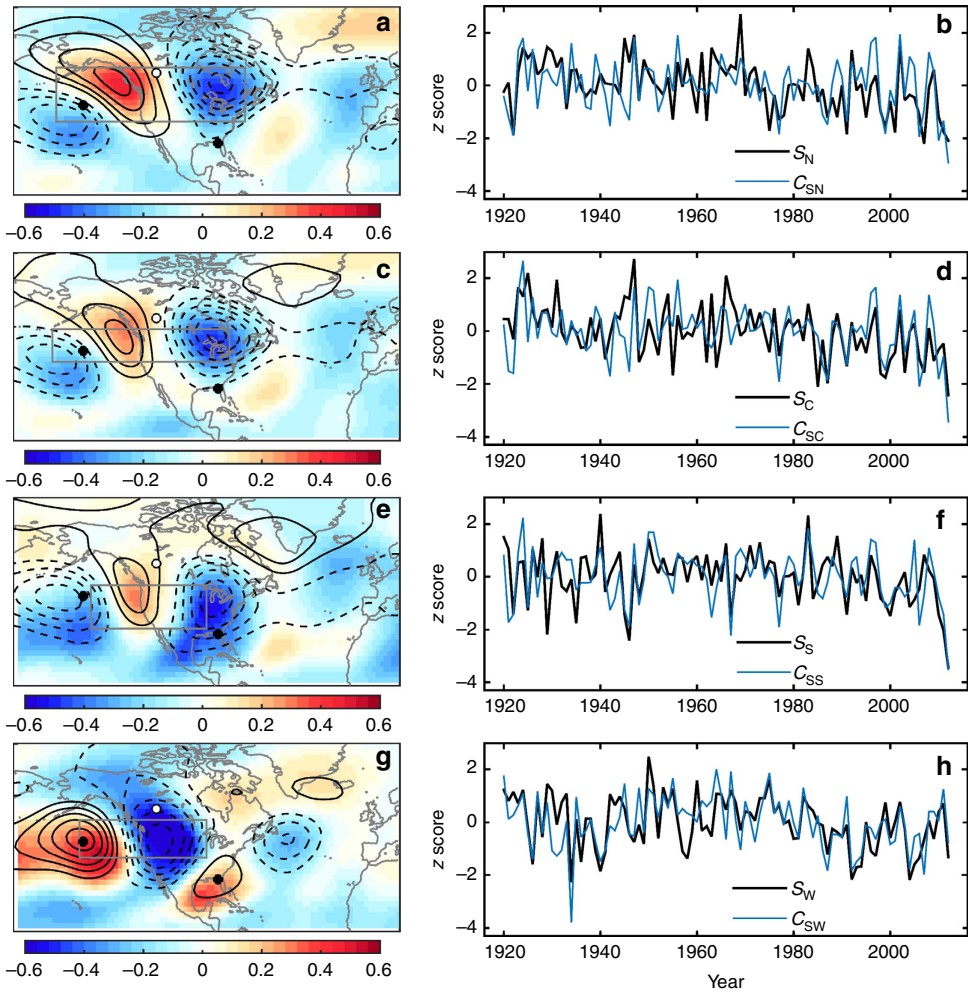

**Figure 3 | Circulation drivers of spring frost timing.** (**a**) Correlation between the day of year with last spring frost in the North region ($S_N$) and mean March–May 500-hPa geopotential height at each grid point ($Z_{MAM}$). The atmospheric teleconnection that captures this $Z_{MAM}$ pattern is referred to as $C_{SN}$, and is the second principal component of $Z_{MAM}$ within the grey box. The $C_{SN}$ spatial pattern is contoured in black at 5 m per standard deviation of the $C_{SN}$ index with negative values dashed and the zero contour suppressed. On all maps, centres of action for the PNA are indicated by small filled white and black circles. (**b**) $S_N$ (black) and the $C_{SN}$ time series (blue). (**c**) Correlation between $Z_{MAM}$ and the day of year with last spring frost in the Central region ($S_C$). The associated circulation mode ($C_{SC}$) is the second principal component of $Z_{MAM}$ within the grey box. (**d**) $S_C$ (black) and the $C_{SC}$ time series (blue). (**e**) Correlation between $Z_{MAM}$ and the day of year with last spring frost in the South region ($S_S$). The associated circulation mode ($C_{SS}$) is the third principal component of $Z_{MAM}$ within the grey box. (**f**) $S_S$ (black) and the $C_{SS}$ time series (blue). (**g**) Correlation between $Z_{MAM}$ and the day of year with last spring frost in the West region ($S_W$). The associated circulation mode ($C_{SW}$) is the first principal component of $Z_{MAM}$ within the grey box. (**h**) $S_W$ (black) and the $C_{SW}$ time series (blue).

with patchy statistical significance (Supplementary Table 1). The strongest relationship was found between $S_W$ and the PNA index, with 29% of the variance explained. The preceding is consistent with the correlation between September and November mean 500-hPa geopotential height ($Z_{SON}$) and $S_W$ indicating a negative PNA-like pattern, although the PNA's centre of action over central Canada is displaced poleward from the strong negative correlation centre over the western U.S. (compare shading to small black and white circles, Fig. 3g).

**Fall frost timing relationship with circulation.** Correlations between the $F$ indices and $Z_{SON}$ revealed circulation patterns that were similar to the patterns found above for the $S$ indices, but with opposite sign (compare maps in Figs 3 and 4). The reversal of sign is logical because an increase in $F$ (that is, later first fall frost) would be associated with ridging, whereas an increase in

$S$ (that is, later last spring frost) would be associated with troughing. For the fall season North region, the tripole circulation pattern found in the correlation between $Z_{SON}$ and $F_N$ (shading, in Fig. 4a) was captured well by the second PC of $Z_{SON}$ within the grey box in Fig. 4a ($C_{FN}$), and the $C_{FN}$ time series accounted for 41% of the variance in $F_N$ (Fig. 4b; Table 1). Partial alignment of the $C_{SC}$ pattern with the PNA's centre of action over the eastern U.S. (compare black contours to small black and white circles, Fig. 4a) provided modest correlation between $C_{SC}$ and the PNA index ($r^2 = 0.21$, Table 1).

For the Central region, the tripole circulation pattern found in the correlation between $Z_{SON}$ and $F_C$ (shading, Fig. 4c) was captured well by the second PC of $Z_{SON}$ within the grey box in Fig. 4c ($C_{FC}$), and the $C_{FC}$ time series accounted for 25% of the variance in $F_C$ (Fig. 4d; Table 1). The $C_{FC}$ pattern is in antiphase with the PNA (compare black contours to small black and white circles, Fig. 4c), and its correlation with the PNA index was not statistically significant (Table 1).

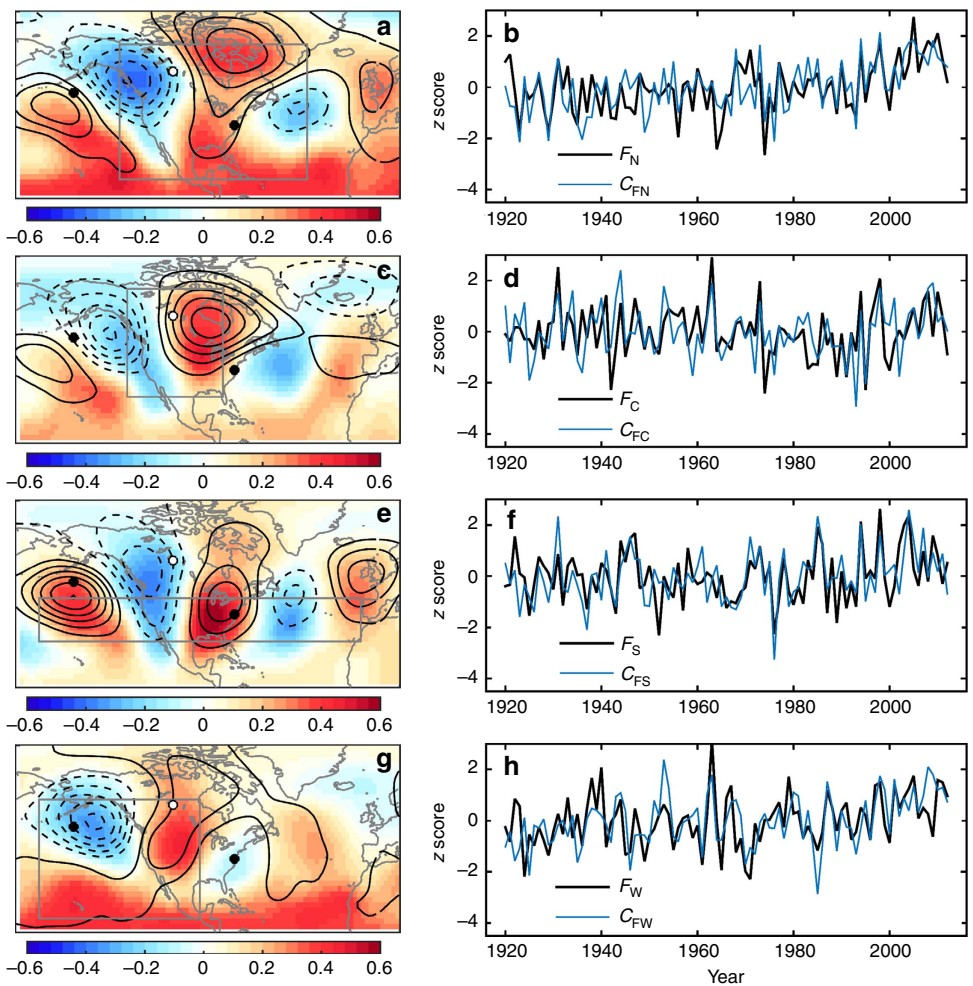

**Figure 4 | Circulation drivers of fall-frost timing.** (**a**) Correlation between the day of year with first fall frost in the North region ($F_N$) and mean September–November 500-hPa geopotential height at each grid point ($Z_{SON}$). The circulation mode that captures this $Z_{SON}$ pattern is denoted $C_{FN}$, and is the second principal component of $Z_{SON}$ within the grey box. The $C_{FN}$ spatial pattern is contoured in black at 5 m per standard deviation of the $C_{FN}$ index with negative values dashed and the zero contour suppressed. On all maps, centres of action for the PNA are indicated by small filled white and black circles. (**b**) $F_N$ (black) and the $C_{FN}$ time series (blue). (**c**) Correlation between $Z_{SON}$ and the day of year with first fall frost in the Central region ($F_C$). The associated circulation mode ($C_{FC}$) is the second principal component of $Z_{SON}$ within the grey box. (**d**) $F_C$ (black) and the $C_{FC}$ time series (blue). (**e**) Correlation between $Z_{SON}$ and the day of year with first all frost in the South region ($F_S$). The associated circulation mode ($C_{FS}$) is the first principal component of $Z_{SON}$ within the grey box. (**f**) $F_S$ (black) and the $C_{FS}$ time series (blue). (**g**) Correlation between $Z_{SON}$ and the day of year with first fall frost in the West region ($F_W$). The associated circulation mode ($C_{FW}$) is the first principal component of $Z_{SON}$ within the grey box. (**h**) $F_W$ (black) and the $C_{FW}$ time series (blue).

For the South region, the circulation anomaly pattern found in the correlation between $Z_{SON}$ and $F_S$ (shading, Fig. 4e) was captured well by the first PC of $Z_{SON}$ within the grey box in Fig. 4e ($C_{FS}$), and the $C_{FS}$ time series accounted for 38% of the variance in $F_S$ (Fig. 4f; Table 1). The $C_{FS}$ pattern aligned reasonably well with the PNA centres of action (compare contours and small black and white circles in Fig. 4e), and $C_{FS}$ was significantly negatively correlated with the PNA index ($r^2 = 0.59$, Table 1).

For the West region, the correlation between $Z_{SON}$ and $F_W$ featured troughing over the Gulf of Alaska indicating an intensified Aleutian low accompanied by downstream ridging over the West region reflecting anomalously warm conditions (shading, Fig. 4g). The spatial pattern of the first PC of $Z_{SON}$ within the grey box in Fig. 4g ($C_{FW}$) captured this correlation dipole structure well (compare black contours to shading, Fig. 4g), and the $C_{FW}$ time series accounted for 29% of the variance in $F_W$ (Fig. 4h; Table 1). The correlation between $F_W$ and

the PNA index was not statistically significant (Table 1) in part because of weak correlations between $Z_{SON}$ and $F_W$ over the PNA's eastern centre of action (compare shading and black contours to small black and white circles, Fig. 4g).

For comparison, we calculated fall-frost timing correlations with indices of the PNA, North Atlantic Oscillation, Atlantic Multidecadal Oscillation and Southern Oscillation, finding correlations smaller than the circulation indices presented above with patchy statistical significance (Supplementary Table 1). The strongest relationship was found between $F_S$ and the PNA index ($r^2 = 0.18$), which is consistent with the correlation pattern between $F_S$ and $Z_{SON}$ (shading, Fig. 4e) being negative-PNA like.

**Relative importance of temperature and circulation.** Circulation indices consistently accounted for more frost timing variance than did hemispheric or regional seasonal temperature, with the distinction between circulation and regional temperature being

smallest in the South and West regions during Spring and in the Central and West regions during fall (Table 1). After linearly removing the effect of the circulation indices, $T_H$ accounted for up to 10% of the residual frost index variance during spring (numbers in parentheses, Table 1). During fall, $T_H$ accounted for negligible additional variance in the frost indices (numbers in parentheses, Table 1) in part because the circulation indices were correlated with $T_H$ (Table 1), suggesting that warming has amplified the circulation patterns, particularly $C_{FN}$ and $C_{FW}$. The regional seasonal temperature indices ($T_R$) accounted for <5% of frost timing variance after linearly removing the effects of the circulation indices (numbers in parentheses, Table 1), which is consistent with the $T_R$ indices being correlated with the circulation indices (Table 1). In particular, the correlation between regional temperature and circulation was very strong in the West region during spring ($T_R$ and $S_W$ have $r^2 = 0.84$, Table 1), consistent with prior studies identifying strong correlations between the leading mode of sea level pressure variability over the northeast Pacific and adjacent coastal warming[19], and also between teleconnections such as the PNA and mountain snowpack[24].

**Sensitivity to number of regions.** The number of regions is given by the number of clusters ($k$) specified in the clustering algorithm (Methods section). To investigate sensitivity to specification of $k$, we repeated the temperature and circulation analysis steps presented above but with different values of $k$ over the range $1 \leq k \leq 10$. For each value of $k$, the analysis steps included partitioning the conterminous U.S. into spring and fall regions based on the clustering and outlier removal method and calculating corresponding regional $S$ and $F$ time series, determining the circulation pattern indices ($C_{xx}$) impacting frost timing in each of the regions via optimization, and calculating the seasonal mean temperature time series ($T_R$) for each region. For the tested range of $k$ during spring and fall, the circulation pattern indices ($C_{xx}$) accounted for more frost-timing variance in each region than did regional seasonal mean temperature ($T_R$) or annual mean hemispheric temperature ($T_H$) (Supplementary Fig. 1). This sensitivity analysis also revealed that frost timing correlation with circulation and temperature tended to decline with increasing $k$ (Supplementary Fig. 2a,b). This decline stemmed in part from the tendency for the variance of the frost timing indices to increase as the average area of the individual cluster regions decreased (Supplementary Fig. 2c).

## Discussion

It is often stated, quite logically, that global warming lengthens the frost-free or growing season[6,25]. The results presented here confirm that the timings of spring and fall frost in four objectively identified regions are significantly correlated with hemispheric-scale temperature indices conventionally used to represent greenhouse gas-driven warming, with the Central region during fall being the one exception. The eight frost timing indices analysed here were also significantly correlated with more local indices of temperature constructed by averaging over the season and region corresponding to each frost index. However, the more profound relationship between frost timing and the atmosphere was dynamical in origin and intermediate between hemispheric and local, consisting of circulation anomaly patterns signalling waveform variations associated with atmospheric teleconnections. The relative importance of dynamically versus radiatively induced temperature variations investigated here may also help to explain contrasting trends in spring onset indices that use growing degree day-based measures which are likely responsive to large-scale warming, versus freeze timing-based measures which are likely responsive to shorter term dynamical variability[26,27].

The atmospheric teleconnections presented here provide circulations conducive to delivery of anomalously warm or cold air masses to the analysis regions, depending on the sign of the circulation index in a particular year. The circulation indices accounted for 25–48% of the frost-timing indices, with the regional temperature indices accounting for negligible amounts of additional variance above and beyond the circulation indices. The frost-driving circulation patterns were broadly PNA-like, meaning that most featured at least three centres of action organized as a wavetrain emanating from the Pacific Ocean. The centres of action associated with the regions over the eastern U.S. were displaced incrementally toward the south as we would expect progressing from the North region into the Central and South regions. The circulation patterns best aligned with the PNA pattern were those driving frost timing in the West region during spring and the South region during fall. The teleconnections impacting frost timing in the North and Central regions during spring, and also the Central region during fall, featured poor alignment and correlation with the PNA, suggesting that these are distinct teleconnections.

Circulation and warming effects are of course intertwined in the observational record because atmospheric circulation modes both affect and are influenced by large-scale air and sea-surface temperature variability[28,29]. The fall circulation patterns identified here are modes of variability (principal components) whose positive polarity signalled anomalous ridging in the respective analysis regions, and all four fall patterns had significant positive correlations with hemispheric-scale temperature suggesting they were amplified by warming. The spring circulation patterns were modes of variability whose positive polarity signalled anomalous troughing in the respective analysis regions, and these modes tended to exhibited weaker negative correlation with hemispheric-scale temperature.

Although frost timing is strongly associated with circulation by theory and case study, there is no *a priori* assurance that circulation patterns can be found that account for more variance than temperature, particularly regional seasonal temperature. The eight patterns identified here are clearly physically plausible drivers of frost timing based on the geographic positioning of their ridging and troughing anomalies, and the importance of these circulation patterns relative to warming speaks to the variance structure of the frost timing indices (that is, trends and low frequency fluctuations we might associate with warming account for less than half the variance of observed frost timing based on spectral decomposition). Based on the sensitivity analysis presented above, the principal finding that circulation consistently accounts for more frost timing variance than hemispheric annual mean or regional seasonal mean temperature does not depend on the number of clusters ($k$) chosen over the test range $1 \leq k \leq 10$.

For future climate change, many studies have investigated how lengthening of the frost-free season could impact ecosystems[30] and agricultural productivity[31]. Pan-continental teleconnections will likely remain important drivers of frost timing variability, even if large-scale warming shifts their effects toward different days of year. Reliable projection of future variations in growing season length thus depends on global climate model fidelity in representing the spatiotemporal properties of this class of teleconnections[32] and how they will respond to large-scale warming.

## Methods

**Frost-timing indices.** For consistency with prior work[5], frost days were defined as having a minimum temperature of 0 °C or lower, the analysis period was 1920–2012, and results were based on 523 stations selected from the Global Historical Climatology Network (indicated by small black symbols in Fig. 1). Our primary variables were the last day of year with spring frost in each year ($S$) and the

first day of year with fall frost in each year ($F$). Analysis regions for spring and fall seasons were objectively identified via $k$-means clustering of $S$ and $F$, respectively, to minimize $f = (1 - R)$, where $R$ denotes distance correlation[33].

**$k$-means clustering.** While various criteria have been developed for assessing $k$-means cluster validity, there is no efficient and universal method for empirically determining the number of clusters ($k$) in a given data set[34]. Here, we selected $k = 4$ based on physical reasoning and the spatial and statistical characteristics of the resultant regions (North, Central, South, West; Fig. 1c,d). To begin with, we sought a set of clusters that was appropriate for the spatial scale of atmospheric teleconnections[35]. The PNA, for example, has two centres of action over the conterminous U.S., suggesting a lower limit of $k = 2$. Then, from analysis of the frost timing data, the decline of $f$ with increasing $k$ had an 'elbow'[36] at $k = 4$ for spring and fall beyond which the decline in $f$ was weaker and nearly linear (Supplementary Fig. 3), indicating utility in increasing $k$ from 2 to 4 with smaller incremental gains thereafter. As another test of reasonableness, selection of $k = 4$ yielded clusters for spring and fall over the eastern U.S. resembling the Köppen–Geiger climate zones[37] (that is, climate zones Dfb, Dfa and Cfa aligned with the North, Central and South regions, respectively)—an alignment missing from alternatives such as $k \in \{2, 3, 5\}$. Specifying $k = 3$ instead of $k = 4$ would have merged the North and Central regions, which are both Df-type Köppen–Geiger climate zones, and would not have changed the fundamental conclusions of the study. Further analysis of the sensitivity to $k$ is presented at the end of results (Section 'Sensitivity to number of regions'). Considering the foregoing reasoning and the strength of the findings presented in results, the specification $k = 4$ provides a useful analysis framework.

To eliminate outlier points present in the initial clustering of $S$ (Supplementary Fig. 4a), each station was assigned the mode of the cluster assignments of its seven nearest neighbours, altering ~9% of the cluster assignments and yielding the refined clusters shown in Supplementary Fig. 4c. The resultant analysis regions for spring are shaded in Fig. 1c. This two-step clustering procedure was then implemented for $F$ (Supplementary Figs. 4b,d), with the resultant anlaysis regions for fall shaded in Fig. 1d. Each station was assigned an area by Dirichlet tessellation for which the associated polygons are evident in Fig. 1a,b, the area-weighted mean $S$ time series were then calculated for each of the four spring regions, and the area-weighted mean $F$ time series were calculated for each of the four fall regions. Because the spring and fall sets of regions shared similar boundaries, we use the same region naming convention for both seasons (North, South, Central and West) as indicated in Fig. 1c,d.

**Climate data.** From the Twentieth Century Reanalysis[38] (20CR) Version 2c, we used ensemble mean 500-hPa geopotential heights ($Z$) and near-surface (model level $\sigma = 0.995$) air temperature ($T$). 20CR data were provided by the NOAA/OAR/ESRL PSD, Boulder, CO, USA, via their web site at http://www.esrl.noaa.gov/psd/.

Monthly indices of the PNA, North Atlantic Oscillation (NAO) and Southern Oscillation Index (SOI) based on 20CR were obtained from the NOAA Earth System Research Laboratory (www.esrl.noaa.gov). Centres of action for the PNA (small black and white circles on maps in Figs 3 and 4) were defined as local extrema of correlation between the PNA index and 500-hPa geopotential height averaged over March through May or September through November. The uncertainty of 20CR increases backward in time as the observation network becomes more sparse, but comparisons with independent radiosonde data indicate that the reanalyses are generally of high quality over the Northern Hemisphere extratropics[38], and we notice no salient shifts or trends in the strength of the correlations analysed here.

The unsmoothed Atlantic Multidecadal Oscillation Index was obtained from the NOAA Earth System Research Laboratory based on Kaplan sea-surface temperatures.[39]

**Statistical analysis.** For the principal components (PC) analyses, the variables were the standardized time series of 500-hPa geopotential height averaged over March through May ($Z_{MAM}$) or September through November ($Z_{SON}$) at each of the $n$ grid point within a region of the Northern Hemisphere defined by a spherical quadrangle. The spherical quadrangle for each PC analysis was optimized via genetic algorithm[40] to maximize correlation between the given frost index and one of the first two PCs of $Z_{MAM}$ or $Z_{SON}$. The optimization was constrained to the Northern Hemisphere and to regions with at least 15° of latitudinal extent and 62.5° of longitudinal extent to avoid identification of correlation pockets over implausibly small regions. To ensure that the algorithm did not hone in on arbitrary domains lacking physical meaning, we visually verified that the loading pattern of each optimized PC aligned with the ridge-trough structures identified in the mapped correlations between geopotential height and frost timing (that is, the contours and shading align well on the maps in Figs 3 and 4). Each time series was of length $m = 93$ years, we assembled these variables into a $m \times n$ matrix, and each PC was an eigenvector of the associated $n \times n$ spatial correlation matrix, meaning it had a spatial vector of length $n$ (for example, black contours within grey box in Fig. 3a) and a corresponding time series of length $m$ (for example, black curve in Fig. 3b). Rather than showing the spatial pattern with arbitrarily scaled units, we

regressed $Z_{MAM}$ or $Z_{SON}$ according to season onto the PC time series[41] so that the units of the map were geopotential height (metres) per s.d. of the standardized PC (for example, black contours in Fig. 3a).

Correlations were tested for significance at the 95% confidence level (indicated using bold font in the tables) using an effective sample size that accounted for lag-one autocorrelation[42].

**Data availability.** The data and reanalysis that support the findings of this study are publicly available online at https://www.ncdc.noaa.gov/oa/climate/ghcn-daily/ and https://www.esrl.noaa.gov/psd/data/20thC_Rean/.

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

## Acknowledgements

C.S. was partially supported by the National Science Foundation under Grant EPS-1208732. We would like to thank Song Feng (Department of Geosciences, University of Arkansas) and three anonymous reviewers for comments that helped improve the manuscript. Any opinions, findings and conclusions or recommendations expressed in this material are those of the authors and do not necessarily reflect the views of the National Science Foundation.

## Author contributions

C.S. and G.J.M. conceived the study and contributed to writing of the manuscript. G.J.M. performed the frost timing analysis and C.S. performed the circulation analysis.

## Additional information

**Competing interests:** The authors declare no competing financial interests.

