## [Peer Review File · Nature Communications]

Reviewers' comments:

Reviewer #1 (Remarks to the Author):

This paper provides an important link between temperature extremes, in this case the frost free season, and atmospheric circulation. The basic conclusion, that the first Fall frost and last Spring frost, are more related to circulation than NH or regional temperatures is not surprising since temperature is tightly coupled to circulation patterns. There are some things that need better explanation and the main question about number of clusters should be addressed.

I am a little dubious of the station clusters (regions). K-means can provide K clusters, why were three chosen? Where other numbers of clusters investigated? I would expect that choosing 5 or even more and performing the same analysis might provide tighter correlations between the regional temperature and the Fall and Spring dates. In a sense this comes down to spatial scale and actually is an interesting question, at what spatial scale does the regional temperature have higher VAR explained than the circulation?

Minor points.

The circulation indices, particularly the PNA should be described as to what kind of circulation patterns they quantify.

The 20th century reanalysis has 56 ensemble members, was the ensemble mean used?

Also uncertainty in the 20th century reanalysis increases farther back in time owing to the increasing sparseness of input data back in time. What are the implications of this?

Reviewer #2 (Remarks to the Author):

The manuscript presents a novel and interesting perspective on the timing of first and last frost across north america, and accordingly will likely be of broad interest to the readership of Nature Climate Change. However, some substantial work is needed to frame the importance of this effort in a more interesting light, and additional text is required to fully justify and explain their methodology. Below are a few recommendations addressing both of these points.

First, the paper does not explain to average reader why we should care about last and first freeze. From an atmospheric dynamics perspective, these are uninteresting and arbitrary thresholds in the variations of weather. There is nothing particularly compelling about studying the circulation anomalies associated with, say, the last day of 5 deg. C in spring, nor is there anything intrinsically interesting about 0 deg C. The reason for studying trends and variability in frost free periods is, of course, that this interval has very significant ecological and agricultural significance. Moreover, a vey large number of studies, and even national and international assessments, point out that one of the (few) benefits of climate change will be a longer "growing season." The study casts doubts on this claim (see comments below), which is an important finding, showing that the frost-free season is driven by dynamic circulation patterns, whereas mean warmth in the spring and fall is more sensitive to mean temperature and/or radiative/greenhouse gas effects. Framing the study to be relevant to a broad range of agroecological applications, as well as a potentially interesting dynamical problem, is key to making this research as high impact as it deserves to be.

Second, in its current form, the manuscript falls short on convincing the reader that the dynamic imprint of circulation trumps climate change, yet this is central to their findings and should be emphasized much more. For example, from the outset, more text could be added to help explain

the difference between “mean warming” and “dynamically modulated” effects on the growing season, which in turn could set up the analysis to weigh in on the current state of knowledge in the field. On a related note, the authors may want to note that they have now provided a more serious dynamical explanation for why several studies have identified a mismatch between the timing of spring onset change (e.g., Schwartz et al., 2006; Allstadt et al., 2015) and the rate of last freeze change (which is slower). Presumably, the implication of the present work is that the thermal-based indices used to identify growing season “start” (e.g., growing degree days) are more sensitive to warming because they integrate temperature over multiple days or weeks, whereas the last freeze/first freeze events are stochastic in nature and hence don’t respond as quickly to mean hemispheric or global temperature.

Third, and most critically, the results hinge on a strange application of cluster analysis and PCA that is not common in climate or meteorological literature. Namely, the authors identify cluster of similar regions, then create time series to represent year-to-year variations in each of these clusters. Additional information on the stopping criterion for these clusters should be provided (e.g., why did they pick 3 and not 5 or 9, or more?). Furthermore, the matching of time series from the clusters to PCS obtained from the geopotential height field is difficult to accept without further justification. As I understand, the authors start with a last freeze or first freeze index from a given region, then start churning through multiple iterations of PCA to identify PC time series that are most similar to the target index by optimizing the spatial domain over which those PCs are calculated. The rationale here is that if the modes of variability governing last/first freeze dates are inherently driven by the atmosphere, they should be present in the EOFs of an appropriate spatial domain; identifying such domains and PCs would verify that the correlation patterns are not merely spurious. This all seems fine, except that I’m left unconvinced that the analysis (as presented) doesn’t just “overfit” the PCs to the last/first freeze indices by honing in an arbitrary spatial domain. Furthermore, as most of the patterns look, in a general sense, “PNA-like” more work connecting their own versions of the PNA to the (more) objectively defined PNA would be very welcome. For example, the findings of Quadrelli and Wallace (2004) suggest most atmospheric modes can project onto the state space of the PNA and NAM, implying that the “modes” identified here can as well. In its current form, however, it is unclear if the patterns and PC time series in geopotential height are intrinsic “modes” of variability, or merely convenient ways to partition the data so that it fits with the last/first freeze indices.

Reviewer #3 (Remarks to the Author):

Comments on manuscript entitled “Observed variations in U.S. frost timing linked to atmospheric circulation patterns” by Strong and McCabe

The authors use objective k-means clustering to identify three regions characterized by different frost timing and reveal the their links to atmospheric circulation anomalies. Crucial to their results is to highlight the relative importance of temperature and circulation patterns in driving frost timing . The results are novel and the paper is well written and organized and in my opinion would interest a broad Nature readership. Therefore I would like to suggest the editor to accept the paper after clarifying some issues.

1. Quality of atmospheric data sets over the North Pacific.

The 20CR is not very good over the North Pacific. There is hardly any information there to be assimilated, especially before the 1950s. The ensemble spread is presumably very large, and the circulation is strongly determined by SSTs and the model’s response to it. And that model can be biased. I think this should be addressed in your paper.

2. The relative importance of temperature and circulation

The authors discuss the relative influence of temperature and circulation by linearly removing the

effect of the circulation indices. This is ok! But Compo and Sardeshmukh (2010) have suggested a more ideal method. So I give this information for your reference.

3. The contribution of circulation to temperature.

The authors discuss the influence of warming on circulation (warming has amplified the circulation patterns). However circulation also has an important influence on regional warming. For example, the PNA circulation pattern has contributed greatly to warming in northwestern USA (Johnstone and Mantua, 2014), as revealed by higher correlation between Tr and Sw than between Csw and Sw (Table 1b and c: the value for Csw/Sw is 0.36 rather than 0.48 reported in the Table). You may need to add some discussion about this.

4. You have mentioned "atmospheric circulation teleconnections" many times in the text. I will suggest you just use "atmospheric circulations" or "atmospheric teleconnections" in your paper.

5. In the last line of P1, ".....over the past century are not well understood.", "are" should be "is" ?

6. In the line 3 of P2, "On the other hand, shifts in mean temperature do not necessarily alter the probability of temperature extremes.....". This may be not the truth. According to Mearns (1984) and Meehl (2000) a small change in the mean temperature causes shifts of the probabilities of extreme temperature events.

7. In the last paragraph of P3, "fall (September through October)" should it be "fall (September through November)"?

8. In Section of Relationship to atmospheric circulation, for the last sentence of the first paragraph, I guess " $r=0.01$ " should be " $r^2=0.01$ " according to what you have reported in Table 1 and Supplementary Table 1. Also, "Table 1f" here should be "Supplementary Table 1f".

9. I am very much confused by the fraction of variance explained for Cxx. What you have reported in the text are different from what you listed in Table 1 and Supplementary Table 1. For example, the Cxx are 48%, 0.36% and 0.36% for Sn, Ss and Sw, respectively in Section of Relationship to atmospheric circulation, but the Cxx are 34%, 0.36% and 0.48% for Sn, Ss and Sw, respectively in the tables.

10. Did you account for time series autocorrelation when you did your statistical tests for the significance? I don't see any information in the methods.

11. Please add line numbers for your text.

References:

1. Compo, G. P., & Sardeshmukh, P. D. (2010). Removing ENSO-related variations from the climate record. *Journal of Climate*, 23(8), 1957-1978.
2. Johnstone, J. A., & Mantua, N. J. (2014). Atmospheric controls on northeast Pacific temperature variability and change, 1900–2012. *Proceedings of the National Academy of Sciences*, 111(40), 14360-14365.
3. Mearns, L. O., Katz, R. W., & Schneider, S. H. (1984). Extreme high-temperature events: changes in their probabilities with changes in mean temperature. *Journal of Climate and Applied Meteorology*, 23(12), 1601-1613.
4. Meehl, G. A., Thomas, K., Easterling, D. R., & Changnon, S. (2000). An introduction to trends in extreme weather and climate events: observations, socioeconomic impacts, terrestrial ecological impacts, and model projections. *Bulletin of the American Meteorological Society*, 81(3), 413.

Reviewer #1 (Remarks to the Author):

1. This paper provides an important link between temperature extremes, in this case the frost free season, and atmospheric circulation. The basic conclusion, that the first Fall frost and last Spring frost, are more related to circulation than NH or regional temperatures is not surprising since temperature is tightly coupled to circulation patterns. There are some things that need better explanation and the main question about number of clusters should be addressed.

We thank the reviewer for the comments and reply to each in turn below, indicating associated improvements to the manuscript.

2. I am a little dubious of the station clusters (regions). K-means can provide K clusters, why were three chosen? Where other numbers of clusters investigated?

We added a new second paragraph to Methods and new Supplementary Figure 1 to present the rationale and quantitative basis for selection of k . Based on these additions, we chose to increase k to 4, noting that this increase did not change the main conclusions of the study. Briefly, $k = 4$ was motivated by the “elbow” plot results in new Supplementary Figure 1. Also, $k = 4$ produced an informative contrast in the associated circulation patterns over the eastern U.S. as now noted in results (the centers of action associated with the regions over the eastern U.S. were displaced incrementally toward the south as we would expect progressing from the North region into the Central and South regions). As an additional check of reasonableness, $k = 4$ yielded regions over the eastern U.S. resembling the Köppen-Geiger climate classification. We also added a new final section to results presenting a sensitivity analysis to k (i.e., the number of regions), referring to new supplemental Figures 3 and 4 and finding that the choice of k did not alter the main conclusions.

3. I would expect that choosing 5 or even more and performing the same analysis might provide tighter correlations between the regional temperature and the Fall and Spring dates. In a sense this comes down to spatial scale and actually is an interesting question, at what spatial scale does the regional temperature have higher VAR explained than the circulation?

Indeed, this is an interesting question about spatial scale. We found that increases in k tended to decrease the overall (i.e., area-weighted mean) correlation between the frost indices and regional seasonal mean temperature and also between the frost indices and the circulation indices, and we present this in new Supplementary Figure 4. Associated

discussion in the new “Sensitivity to number of regions” results section notes that correlation declined with increasing k (i.e., increasing number of regions) in part because frost timing variance tended to increase as the average area of the regions was reduced (Supplemental Figure 4c).

Minor points.

4. The circulation indices, particularly the PNA should be described as to what kind of circulation patterns they quantify.

We added text to the Introduction referencing Leathers et al. (1991) to note that the PNA is characterized in its positive polarity by troughing over the eastern Pacific, ridging over the Rocky Mountains, and troughing over eastern North. We also show the centers of action of the PNA (defined in Methods) for spring and fall by small white and black circles in Figures 3 and 4, and discuss in results the degree of alignment for each circulation pattern.

5. The 20th century reanalysis has 56 ensemble members, was the ensemble mean used?

We used the ensemble mean and now note this in Methods.

6. Also uncertainty in the 20th century reanalysis increases farther back in time owing to the increasing sparseness of input data back in time. What are the implications of this?

We added text to Methods to note that the uncertainty of 20CR increases backward in time as the observation network becomes sparser. We also referred to Compo et al. (2011) to note that comparisons with independent radiosonde data indicate that the reanalyses are generally of high quality over the Northern Hemisphere extratropics, and we note that we find no salient decreases or shifts in the strength of the correlations analyzed here.

Reviewer #2 (Remarks to the Author):

1. The manuscript presents a novel and interesting perspective on the timing of first and last frost across north america, and accordingly will likely be of broad interest to the readership of Nature Climate Change. However, some substantial work is needed to frame the importance of this effort in a more interesting light, and additional text is required to fully justify and explain their methodology. Below are a few recommendations addressing both of these points.

We thank the reviewer for the comments and reply to each in turn below, indicating associated improvements to the manuscript.

2. First, the paper does not explain to average reader why we should care about last and first freeze. From an atmospheric dynamics perspective, these are uninteresting and arbitrary thresholds in the variations of weather. There is nothing particularly compelling about studying the circulation anomalies associated with, say, the last day of 5 deg. C in spring, nor is there anything intrinsically interesting about 0 deg C. The reason for studying trends and variability in frost free periods is, of course, that this interval has very significant ecological and agricultural significance. Moreover, a vey large number of studies, and even national and international assessments, point out that one of the (few) benefits of climate change will be a longer “growing season.” The study casts doubts on this claim (see comments below), which is an important finding, showing that the frost-free season is driven by dynamic circulation patterns, whereas mean warmth in the spring and fall is more sensitive to mean temperature and/or radiative/greenhouse gas effects. Framing the study to be relevant to a broad range of agroecological applications, as well as a potentially interesting dynamical problem, is key to making this research as high impact as it deserves to be.

We appreciate the suggestion to highlight this high impact, and we tightened up the abstract to make room for text highlighting that “These teleconnections appear responsive to historical warming ...” and “Reliable projections of future variations in growing season length depend on the fidelity of these circulation patterns in global climate models.” We also add text to the final paragraph of Summary and Discussion with references to Chapters 4 and 7 of the IPCC AR5 report. We hesitate to explicitly cast doubt on studies of future growing season length because they should be reliable as long as the models they are based on capture these teleconnection effects with fidelity, regardless of whether the analysts are aware of the underlying teleconnection dynamics. An important component of the problem is thus spatial and temporal biases in how global climate models represent teleconnections, and our new text in Summary and Discussion emphasizes this with a reference to the assessment by Stoner et al. (2009).

3. Second, in its current form, the manuscript falls short on convincing the reader that the dynamic imprint of circulation trumps climate change, yet this is central to their findings and should be emphasized much more. For example, from the outset, more text could be added to help explain the difference between “mean warming” and “dynamically modulated” effects on the growing season, which in turn could set up the analysis to weigh in on the current state of knowledge in the field.

We split the second paragraph of the Introduction into new paragraphs 2 and 3 to set up the contrast in radiatively-induced versus dynamically-induced temperature effects. Paragraph 2 now notes evidence for both in driving frost timing, incorporating references to Wallace et al. (1995) and Guan et al. (2015) on intertwining of the signals in the historical record. Paragraph 3 reports that prior studies focused on certain regions and periods find frost timing correlation with teleconnections, but recent work extending the analysis to a century of data with coverage over the conterminous U.S. found statistically significant linkages to only the AMO, leaving the roles and relative importance of large-scale warming and dynamically-induced atmospheric circulation effects unclear.

4. On a related note, the authors may want to note that they have now provided a more serious dynamical explanation for why several studies have identified a mismatch between the timing of spring onset change (e.g., Schwartz et al., 2006; Allstadt et al., 2015) and the rate of last freeze change (which is slower). Presumably, the implication of the present work is that the thermal-based indices used to identify growing season “start” (e.g., growing degree days) are more sensitive to warming because they integrate temperature over multiple days or weeks, whereas the last freeze/first freeze events are stochastic in nature and hence don’t respond as quickly to mean hemispheric or global temperature.

We added a sentence to the end of the first paragraph of Summary and Discussion referring to these two studies and stating that the relative importance of dynamically- versus radiatively-induced temperature variations may help to explain contrasting trends in spring onset indices that use growing degree day-based measures which are likely responsive to large-scale warming, versus freeze timing-based measures which are likely responsive to shorter term dynamical variability.

5. Third, and most critically, the results hinge on a strange application of cluster analysis and PCA that is not common in climate or meteorological literature. Namely, the authors identify cluster of similar regions, then create time series to represent year-to-year variations in each of these clusters. Additional information on the stopping criterion for these clusters should be provided (e.g., why did they pick 3 and not 5 or 9, or more?).

As noted in response to a similar comment from Reviewer #1, we added a new second paragraph to Methods and new Supplementary Figure 1 to present the rationale and quantitative basis for selection of k . Based on these additions, we chose to increase k to 4, noting that this increase did not change the main conclusions of the study. Briefly, $k = 4$ was motivated by the “elbow” plot results in new Supplementary Figure 1. Also, $k = 4$ produced an informative contrast in the associated circulation patterns over the eastern U.S. as now noted in results (the centers of action associated with the regions over the eastern U.S. were displaced incrementally toward the south as we would expect progressing from the North region into the Central and South regions). As an additional check of reasonableness, $k = 4$ yielded regions over the eastern U.S. resembling the Köppen-Geiger climate classification. We also added a new final section to results presenting a sensitivity analysis to k (i.e., the number of regions), referring to new supplemental Figures 3 and 4 and finding that the choice of k did not alter the main conclusions.

6. Furthermore, the matching of time series from the clusters to PCS obtained from the geopotential height field is difficult to accept without further justification. As I understand, the authors start with a last freeze or first freeze index from a given region, then start churning through multiple iterations of PCA to identify PC time series that are most similar to the target index by optimizing the spatial domain over which those PCs are calculated. The rationale here is that if the modes of variability governing last/first freeze dates are inherently driven by the atmosphere, they should be present in the EOFs of an appropriate spatial domain; identifying such domains and PCs would verify that the correlation patterns are not merely spurious. This all seems fine, except that I’m left unconvinced that the analysis (as presented) doesn’t just “overfit” the PCs to the last/first freeze indices by honing in on an arbitrary spatial domain.

We ensured that the optimization algorithm did not hone in on an arbitrary domain by verifying that the PC loading pattern aligned with the ridge-trough patterns found in the mapped correlations between frost timing and geopotential height (i.e., the black contours align with the shading on the maps in Figures 3 and 4). We see that this step was underemphasized in the submitted draft, and to remedy this, we added the following sentence to Methods: “To ensure that the algorithm did not hone in on arbitrary domains lacking physical meaning, we visually verified that the loading pattern of each optimized PC aligned with the ridge-trough structures identified in the mapped correlation between geopotential height and frost timing (i.e., the contours and shading align well on the maps in Figures 3 and 4).” We also added text to emphasize this point when presenting the first optimization for spring in the North region (section “Spring frost timing relationship with circulation”).

7. Furthermore, as most of the patterns look, in a general sense, “PNA-like” more work connecting their own versions of the PNA to the (more) objectively defined PNA would be very welcome. For example, the findings of Quadrelli and Wallace (2004) suggest most atmospheric modes can project onto the state space of the PNA and NAM, implying that the “modes” identified here can as well. In its current form, however, it is unclear if the patterns and PC time series in geopotential height are intrinsic “modes” of variability, or merely convenient ways to partition the data so that it fits with the last/first freeze indices.

We made several additions to clarify the relationship between the PNA and the teleconnections presented in the study. First, we added text to the Introduction referencing Leathers et al. (1991) to note that the PNA is characterized in its positive polarity by troughing over the eastern Pacific, ridging over the Rocky Mountains, and troughing over eastern North. We added the centers of action of the PNA to Figures 3 and 4 for visual comparison, and noted their derivation in Methods. In results, we describe how the teleconnections affecting some of the regions (e.g., North region during spring) are in antiphase with the PNA, while those affecting other regions were better aligned (e.g., West region during spring), and we note how this corresponds to the correlations between the circulation indices and the PNA (now added as row f in Table 1). Finally, we expanded the second paragraph of Summary and Discussion to further compare and contrast the circulation patterns and the PNA.

Reviewer #3 (Remarks to the Author):

Comments on manuscript entitled "Observed variations in U.S. frost timing linked to atmospheric circulation patterns" by Strong and McCabe

The authors use objective k-means clustering to identify three regions characterized by different frost timing and reveal their links to atmospheric circulation anomalies. Crucial to their results is to highlight the relative importance of temperature and circulation patterns in driving frost timing. The results are novel and the paper is well written and organized and in my opinion would interest a broad Nature readership. Therefore I would like to suggest the editor to accept the paper after clarifying some issues.

We thank the reviewer for the comments and reply to each in turn below, indicating associated improvements to the manuscript.

1. Quality of atmospheric data sets over the North Pacific.

The 20CR is not very good over the North Pacific. There is hardly any information there to be assimilated, especially before the 1950s. The ensemble spread is presumably very large, and the circulation is strongly determined by SSTs and the model's response to it. And that model can be biased. I think this should be addressed in your paper.

As noted in reply to a similar comment from Reviewer #1, we added text to Methods to note that the uncertainty of 20CR increases backward in time as the observation network becomes sparser. We also referred to Compo et al. (2011) to note that comparisons with independent radiosonde data indicate that the reanalyses are generally of high quality over the Northern Hemisphere extratropics, and we note that we find no salient decreases or shifts in the strength of the correlations analyzed here.

2. The relative importance of temperature and circulation

The authors discuss the relative influence of temperature and circulation by linearly removing the effect of the circulation indices. This is ok! But Compo and Sardeshmukh (2010) have suggested a more ideal method. So I give this information for your reference.

We thank the reviewer for pointing us to this alternative to linearly removing effects, and will take the time to understand and incorporate these methods into our future research.

3. The contribution of circulation to temperature.

The authors discuss the influence of warming on circulation (warming has amplified the circulation patterns). However circulation also has an important influence on regional warming. For example, the PNA circulation pattern has contributed greatly to warming in northwestern USA (Johnstone and Mantua,

2014), as revealed by higher correlation between T_r and S_w than between C_{sw} and S_w (Table 1b and c: the value for C_{sw}/S_w is 0.36 rather than 0.48 reported in the Table). You may need to add some discussion about this.

In the section "Roles of temperature and circulation," we added text with reference to Johnstone and Mantua (2014) and Abatzoglou (2011) to contextualize the strong control of circulation on regional temperature indicated in our results, particularly in the West region. Regarding the correlations, please note that the 0.48 in the table was correct and the 0.36 in the main text was incorrect.

4. You have mentioned "atmospheric circulation teleconnections" many times in the text. I will suggest you just use "atmospheric circulations" or "atmospheric teleconnections" in your paper.

We eliminated all instances of "atmospheric circulation teleconnections" as suggested.

5. In the last line of P1, ".....over the past century are not well understood.", "are" should be "is" ?

Yes, we changed "are" to "is."

6. In the line 3 of P2, "On the other hand, shifts in mean temperature do not necessarily alter the probability of temperature extremes.....". This may be not the truth. According to Mearns (1984) and Meehl (2000) a small change in the mean temperature causes shifts of the probabilities of extreme temperature events.

This is a good point, and we do not explicitly consider distributions in the analysis presented here. Considering this comment and our restructuring the content of the paragraph around the concepts of radiatively-induced versus dynamically-induced warming in response to comment 3 from Reviewer 2, we removed this sentence from the manuscript.

7. In the last paragraph of P3, "fall (September through October)" should it be "fall (September through November)"?

Yes, all fall analyses were September through November, and this has been corrected.

8. In Section of Relationship to atmospheric circulation, for the last sentence of the first paragraph, I guess " $r=0.01$ " should be " $r^2=0.01$ " according to what you have reported in Table 1 and Supplementary Table 1. Also, "Table 1f" here should be "Supplementary Table 1f".

Yes, that should have been r^2 rather than r . The sentence now just refers to the table row without reporting the value because this correlation is not significant at the 95% confidence level.

9. I am very much confused by the fraction of variance explained for Cxx. What you have reported in the text are different from what you listed in Table 1 and Supplementary Table 1. For example, the Cxx are 48%, 0.36% and 0.36% for Sn, Ss and Sw, respectively in Section of Relationship to atmospheric circulation, but the Cxx are 34%, 0.36% and 0.48% for Sn, Ss and Sw, respectively in the tables.

The tabulated values were correct, and we have corrected discrepancies between the text and table. Also note that all values have been updated to accommodate the increase to $k = 4$ clusters in response to Reviewer 1 Comment 2 and Reviewer 2 Comment 5.

10. Did you account for time series autocorrelation when you did your statistical tests for the significance? I don't see any information in the methods.

We added a final paragraph to Methods to clarify that correlations were tested for significance at the 95% confidence level (indicated using bold font in the tables) using an effective sample size that accounted for lag-one autocorrelation. The referenced paper (Bretherton et al., 1999) is focused on spatial series, but provides a convenient review of methods appropriate for time series as well (Section 5).

11. Please add line numbers for your text.

Line numbers have been added.

References:

1. Compo, G. P., & Sardeshmukh, P. D. (2010). Removing ENSO-related variations from the climate record. *Journal of Climate*, 23(8), 1957-1978.
2. Johnstone, J. A., & Mantua, N. J. (2014). Atmospheric controls on northeast Pacific temperature variability and change, 1900–2012. *Proceedings of the National Academy of Sciences*, 111(40), 14360-14365.
3. Mearns, L. O., Katz, R. W., & Schneider, S. H. (1984). Extreme high-temperature events: changes in their probabilities with changes in mean temperature. *Journal of Climate and Applied Meteorology*, 23(12), 1601-1613.
4. Meehl, G. A., Thomas, K., Easterling, D. R., & Changnon, S. (2000). An introduction to trends in extreme weather and climate events: observations, socioeconomic impacts, terrestrial ecological impacts, and model projections. *Bulletin of the American Meteorological Society*, 81(3), 413.

REVIEWERS' COMMENTS:

Reviewer #1 (Remarks to the Author):
only comments to the editor

Reviewer #2 (Remarks to the Author):

The revised manuscript is much stronger and more clear, and I find it ready for publication. It will likely be of interest to the broader readership of nature communications because it addresses a problem with both high ecological and agricultural relevance (frost free periods, or conversely, frost damage in spring or fall). Moreover, the distinction highlighted by the authors between "radiative" and "dynamic" effects on frost events will likely be of broader interest to researchers studying climate and atmospheric dynamics.

Reviewer #3 (Remarks to the Author):

The authors have responded well to my original critique and suggested changes. In particular, the authors have added a sensitivity analysis to their results by selecting different numbers of regions, and demonstrate that their results are robust. I recommend that the paper be published as is now.